# Knowledge-Sharing Practices Among Dentists, Pharmacists, and Allied Health Professionals: A Cross-Sectional Study in Eastern Cape Public Hospitals, South Africa

**DOI:** 10.3390/ijerph23010066

**Published:** 2025-12-31

**Authors:** Nombulelo Chitha, Linda Sobekwa, Ziyanda Ngcobo, Ruth Tshabalala, Ntiyiso V. Khosa, Onke R. Mnyaka

**Affiliations:** 1Independent Researcher, Pretoria 0181, South Africa; buli.chitha@gmail.com; 2School of Public Health, Faculty of Medicine and Health Sciences, Walter Sisulu University, Sisson Street, Mthatha 5117, South Africa; lsobekwa@wsu.ac.za (L.S.); zngcobo@wsu.ac.za (Z.N.); rtshabalala@wsu.ac.za (R.T.); nkhosa@wsu.ac.za (N.V.K.); 3WSU Institute for Clinical Governance and Healthcare Administration, Faculty of Medicine and Health Sciences, Walter Sisulu University, East London 5200, South Africa; 4WSU Global Centre for Human Resources for Health Intelligence, Faculty of Medicine and Health Sciences, Walter Sisulu University, East London 5200, South Africa

**Keywords:** knowledge-sharing, knowledge exchange, dentist, pharmacist, allied health professional

## Abstract

Knowledge-sharing is a deliberate exchange of information to enhance accessibility and reuse which is critical for improving healthcare delivery. This study assessed knowledge-sharing practices among dentists, pharmacists, and allied health professionals (AHPs) in nine public hospitals in South Africa’s Eastern Cape Province. A cross-sectional survey was conducted using purposive and stratified random sampling to recruit 99 participants. Data were collected via a validated questionnaire and analysed with SPSS v.22.0 using descriptive statistics. Respondents were predominantly female (77.6%) and aged 21–35 years (63.6%); AHPs comprised 65.7% of the sample. The results show a statistically significant association between profession and encouragement to adopt a global perspective (*p* = 0.017), while significant differences were observed between profession and encouragement to seek inter-team solutions (*p* = 0.020), and access to leadership-driven opportunities for interdisciplinary knowledge-sharing (*p* = 0.016). Despite observable patterns in the descriptive results, no other statistically significant differences by profession were observed for all other items. Collaboration with external communities and leadership-driven knowledge-sharing opportunities were also highest among dentists but limited overall. Adoption of information systems for knowledge exchange was low, particularly among pharmacists and AHPs. Participation in professional development and recognition of long-term knowledge-sharing strategies followed similar patterns. These findings highlight the need to strengthen leadership-driven opportunities for interdisciplinary knowledge-sharing and to develop targeted interventions to address specific gaps between professions.

## 1. Introduction

In a healthcare environment, knowledge-sharing and continuous learning are not just important, they are the backbone for improving patient outcomes, driving innovation, and supporting the professional growth of healthcare providers [1,2]. Knowledge-sharing explains and distributes up-to-date health information to personnel, policymakers, and other sectors through interactive communication channels, a noble and vital task [3]. It involves intentionally making information accessible and reusable for others [4]. The knowledge-sharing habits of healthcare professionals are not just important, however; they are the key to the success of healthcare organisations such as hospitals [3], offering them a long-term competitive edge [5]. The healthcare sector heavily relies on its resources, with knowledge being the most distinct and crucial for maintaining a competitive advantage [6].

Knowledge can be shared by disseminating documented information or experience-based insights (tacit knowledge) [7]. Although tacit knowledge is crucial in clinical decision-making and problem-solving, it often remains unshared due to significant barriers [8]. These include time constraints, hierarchical structures, and the lack of formal knowledge-sharing systems [7]. The persistent reluctance by employees to share knowledge is also linked to organisational culture [6]. Therefore, developing various learning strategies to enhance knowledge acquisition and sharing is imperative [9].

Despite the acknowledgement of knowledge-sharing among different health systems globally, it remains poorly practised [5], especially in hospitals in resource-scare countries [3]. A study conducted in China among referral healthcare services indicated that 61% of hospital doctors rated knowledge-sharing as a very poor practice [5]. Hospitals play a critical role in this process as a significant link in a complex continuum of care [10]. Hospitals are knowledge-intensive environments where strategic use of information is essential for strengthening the health system. They involve a diverse group of professionals with expertise in various disciplines who must stay informed about continual updates in relevant technologies and fields of knowledge [1]. In hospitals, establishing a learning culture characterised by a shared commitment to learning, knowledge-sharing, and continuous improvement is important [11]. A positive learning environment fosters ongoing skill development among health professionals, encourages the adoption of best practices, and promotes knowledge exchange.

A positive organisational culture, strong organisational commitment, and active organisational citizenship behaviour are essential for fostering an enabling environment for sharing knowledge [12]. This culture supports professional development and enhances collaborative practices [11]. While it is common in hospitals that physicians and nurses primarily render clinical services, it is crucial to recognise that allied healthcare professionals (AHPs), dentists and pharmacists also play a significant role [13]. They often must leverage their expertise to collaborate with professionals from various fields and to solve problems cooperatively [1]. AHPs provide diagnostic, technical, therapeutic, and clinical support, as highlighted by Weng et al. [13]. Dentists provide preventive care, perform restorative procedures, and offer emergency dental services [14]. They also collaborate with other medical professionals to address oral health issues that can affect overall health, especially in patients with chronic diseases or those undergoing treatments such as chemotherapy [14]. Pharmacists, on the other hand, ensure the safe and effective use of medications [15]. They manage patients’ medication regimens and provide essential counselling on proper medication use and potential side effects. Pharmacists also collaborate with other healthcare team members to optimise patient care and ensure the quality and safety of medications through proper storage and handling [15].

Knowledge-sharing at individual and organizational levels yields inherent [3]. Research demonstrates that hospitals with strong learning cultures and effective knowledge-sharing systems can deliver high-quality care and respond innovatively to emerging health challenges [16]. Effective knowledge-sharing practices can significantly enhance evidence-based clinical decision-making [17]. Various individual (e.g., self-efficacy, motivation, trust), organisational (e.g., leadership support, collaborative culture, workload) and technological factors (e.g., usability of ICT systems, access to digital tools) can hinder or facilitate effective knowledge-sharing and learning culture [1,18]. Support from top management, organizational rewards, and technological resources are linked to knowledge-sharing practices [5]. Other factors that can influence knowledge-sharing practices include education level, experience, awareness, willingness, teamwork, availability of health information resources, communication mechanisms, internet access, and computer literacy [19]. Another study in Ethiopia [3] suggests that factors that negatively affect knowledge-sharing practices include individual factors and organisational and resource-related issues [3]. In order for hospitals to create an environment that supports continuous knowledge-sharing among healthcare professionals, they must understand these dynamics.

Health professionals from resource-limited countries are known for their limited information-sharing practices. Studies conducted in Ethiopia and Malaysia reported poor knowledge-sharing practices among health professionals [3]. These knowledge-sharing practices can be attributed to education level, experience, awareness, willingness, teamwork, availability of health information resources, communication mechanisms, internet access, and computer literacy [19]. It is common for healthcare workers to rely on the knowledge they acquired in school without accessing knowledge from their colleagues [19].

Healthcare delivery is becoming increasingly complex, necessitating collaboration among professionals from diverse disciplines to ensure timely and effective clinical decision-making. Inadequate knowledge-sharing in such environments can result in fragmented care, duplication of tasks, preventable errors, and suboptimal patient outcomes.

These challenges are exacerbated in South Africa’s Eastern Cape (EC) province, a resource-limited province in South Africa, where healthcare access is severely constrained by structural inequities between rural and urban populations [17]. Specifically, hospitals in the EC province are confronted with challenges like limited access to training resources, limited digital infrastructure, staff shortages, and high patient volumes [20]. Moreover, rural communities face significant barriers, including long travel distances to urban-based tertiary or quaternary facilities, limited transportation infrastructure, and financial constraints that make private care unattainable [21]. These systemic gaps often result in delayed treatment, increased morbidity, and poorer health outcomes for rural populations [21]. Also, resource limitations, such as shortages of skilled personnel, inadequate digital infrastructure, and insufficient training opportunities, compound the difficulty of delivering coordinated, high-quality care.

The advantages of knowledge-sharing include advancements in patient care, enhanced well-being, and reduced medical errors [22,23]. Therefore, addressing these disparities requires not only improving physical access but also fostering robust knowledge-sharing practices among healthcare professionals to enhance collaboration, optimize resource use, and ensure equitable service delivery across rural and urban settings. Despite the known benefits of knowledge-sharing, there is a scarcity of research focusing on knowledge-sharing practices among AHPs, dentists, and pharmacists. This research study assessed the knowledge-sharing practices of AHPs, dentists, and pharmacists in nine EC province public hospitals.

## 2. Materials and Methods

### 2.1. Study Design

A quantitative cross-sectional survey design was followed in this study to leverage its ability to collect data for different variables at a single point in time. This study forms part of a larger study titled “Exploring the use of hospitals as a learning organisation in South Africa’s EC province” [24]. This study assessed the knowledge-sharing practices of dentists, pharmacists and AHPs.

### 2.2. Study Setting

This study was conducted in nine public hospitals across four district municipalities (Alfred Ndzo, Amathole, Chris Hani and OR Tambo) and one metropolitan municipality (Buffalo City) in the EC province, South Africa [24] The EC is one of the nine provinces in South Africa, which ranks as the second-largest province in the country by land area and has the fourth-largest population in South Africa, with about 7 million people living in the province [25]. However, it is also one of the two poorest provinces in the country [26]; about 70% of the population lives in poverty and resides in rural areas [26].

### 2.3. Population and Sampling

Public hospitals in the EC province constitute the population for this study. There are 117 public hospitals in the EC province, national central hospital (1), provincial tertiary hospitals (3), regional hospitals (5), district hospitals (65) and specialised hospitals (18) [23]. Nine hospitals were purposively sampled based on their involvement in training Walter Sisulu University’s students. Five district hospitals were selected (All Saints, Bisho, Butterworth, Dr Malizo Mpehle, Madzikane KaZulu), two regional hospitals (Frontier and Mthatha regional) and two tertiary hospitals (Frere and Cecilia Makiwane). Stratified random sampling was used to select the participants. The sampling approach used in this study is described in detail by Chitha et al. [24]. A total of 99 health professionals, AHPs (65), dentists (11) and pharmacists (23) were recruited to participate in the study.

### 2.4. Inclusion and Exclusion

Several considerations were made before participants were included in this study. Participants were eligible if they were registered dentists, pharmacists, or allied health professionals currently employed in the study hospitals and providing clinical or direct patient-related services; were aged 18 years or older; have been employed at the study hospital for at least six months to ensure familiarity with local practices and were willing to provide written informed consent. While administrative staff with no clinical duties, undergraduate students, staff on extended leave (more than 3 months) during data collection, and those with survey responses with excessive missing data (>20% on key items) were not considered for inclusion in this study.

### 2.5. Development and Validation of the Data Collection Instrument

The questionnaire was developed through literature review and established theoretical frameworks to ensure conceptual coverage. Content validity was assessed by two senior academic researchers, an organizational behaviour specialist and a public health medicine specialist, who evaluated each item for relevance, clarity, and comprehensiveness. Following content validation, the questionnaire was piloted with 10 individuals from the target population. Internal consistency of the nine-item knowledge-sharing scale was assessed using Cronbach’s alpha coefficient, with values ≥ 0.70 considered acceptable for reliability [27,28]. The scale demonstrated excellent reliability (α = 0.88), exceeding the recommended threshold of 0.70 for research instruments.

### 2.6. Data Collection

Recruitment of participants and data collection commenced on the 1 June 2022 and was completed on the 31 December 2022. Repeated visits were done until completion of data collection in all the study sites. Specifically, data was collected at Cecilia Makiwane and Frere hospitals, simultaneously from on 1 to 30 June 2022; Bisho, Butterworth, Frontier and All Saints hospitals, 18 July 2022 to 31 August 2022; Madzikane KaZulu and Dr. Malizo Mpehle hospitals, 5–22 September 2022; and Mthatha Regional hospital, 26 September 2022 to 31 December 2022.

Data were collected through a structured self-administered questionnaire. The questionnaire was developed by the research team and reviewed and validated by the principal investigator, who is an expert in the field. The validation process by an expert ensured the reliability of the questionnaire. The questionnaire asked questions on demographic characteristics, experience-based advising, discussion of learning expectations, learning as a recruitment value, individualised learning plans and partnership for capacity development in the selected hospitals. Printed copies of the data collection instruments were handed out by the research team for completion by all participants who consented to participate. The data collection team endeavoured to accommodate all the prospective participants, including those who work on nightshifts or weekends, through repeated visits to collect data. To ensure that all questions were responded to, the data collection team verified all the questionnaires in the presence of the participants.

### 2.7. Data Management and Analysis

Only selected research team members are permitted to access the collected data. All the hard copy files are stored in an access-controlled data storage room in lockable filing cabinets. Soft copy data is kept on a password-encrypted online storage system. In order to maintain the anonymity of the participants, data were deidentified through coding before capturing the information into a Microsoft Excel spreadsheet. Ms Excel validation settings were used to prevent duplication and other potential errors during data capturing.

Data were analysed using Stata Statistical Software (Version 18; StataCorp LCC). The Shapiro–Wilk test was used to assess the normality of numerical data. Numerical variables (age) are presented using the median, interquartile range (IQR), minimum and maximum, as they were not normally distributed. Tables are used to summarise and present categorical variables. A Chi-square test assessed the association between categorical variables, with significance set at 5%.

### 2.8. Ethical Considerations

This study obtained ethical approval from the University of Witwatersrand’s Human Research Ethics Committee (Ref no: R14/49). Approval to access the selected study sites was obtained from the Eastern Cape Provincial Health Research Committees (PHRC) (Ref no: EC_202108_011). While conducting this study, the researchers conformed to the four ethical principles of autonomy, beneficence, non-maleficence, and justice. This ensured the voluntary participation of all participants by obtaining a signed informed consent form before participating in this study, ensuring anonymity and confidentiality of the information relating to participants.

## 3. Results

The demographic characteristics of the 99 hospital-based health professionals surveyed, comprising dentists (11.1%), pharmacists (23.2%), and AHPs (65.7%) are presented in Table 1. The median age of participants was 32 years (IQR: 13), with ages ranging from 21 to 62 years.

AHPs constituted the largest proportion of respondents across all age groups. They represented 69.8% of participants aged 21–35 years, 47.6% in the 36–45-year group, and remained the majority in older cohorts, although dentists and pharmacists showed relatively higher proportions in these groups.

Regarding Sex of the participants, females represented most respondents overall (76.8%), with the highest proportion found among AHPs (71.1%), followed by pharmacists (19.7%) and dentists (9.2%). Male participants accounted for only 23.2% of the sample, again with AHPs forming the largest share.

Participants were drawn from nine hospitals across the EC province, of which AHPs were in majority across the hospitals. Frere hospital (23.2%) accounted for most participants, followed by Cecilia Makiwane hospital (22.2%), then Mthatha Regional Hospital (18.2%). Dentists and pharmacists were generally more evenly distributed across the hospitals; however, certain facilities, such as Bisho and Butterworth, had very few or no dentists represented.

Table 2 presents the distribution of responses across professions regarding systems that encourage knowledge-sharing and the extent to which departmental theories are promoted in practice including: encouraging global thinking; seeking answers across teams; engaging with the outside community; establishing social structures such as communities of practice; fostering a daily knowledge sharing culture; providing leadership driven opportunities for sharing; adopting information systems to support information exchange; encouraging staff returning from training to share new learning; and establishing long term knowledge sharing strategies. Across all professional categories most respondents reported that their departments frequently (“often” or “always”) promote these behaviours; however, significant differences were observed for three specific practices.

Encouraging staff to think from a global perspective

The results reveal statistically significant association between profession and perceptions that the department encourages staff to think from a global perspective (*p* = 0.017). 36.4% of dentists believe the department promotes a global perspective; the remaining participants indicated that this encouragement occurs often (27.3%), seldom (27.3%) or never (9.1%). Most (56.5%) of pharmacists perceived that the department rarely promotes a global perspective, while 26.1% indicated it never does. On a contrary, 13.0% felt encouraged often and only 4.3% indicated it always does. Among AHPs, 53.9% reported seldom encountering this encouragement, while 29.7% reported that global thinking is often encouraged, whereas 9.2% felt it is never promoted.

Encouraging staff to seek answers from other teams when solving problems

There were also statistically significant differences observed by profession concerning whether departments encourage staff to seek answers from other teams when solving problems (*p* = 0.020). While 45.5% and 36.4% of dentists believed that the department always or often encourages seeking answers across teams when solving problems, and 18.2% did not; mixed responses were observed from pharmacists, 56.5% indicating seldom, 21.7% often, 13.0% never, and 8.7% always; while majority of AHPs, stated seldom (44.6%) or often (38.5%), with fewer indicating always (10.8%) or never (6.2%).

Leadership driven opportunities for knowledge sharing

Table 2 shows statistically significant differences among professions regarding their perceptions of whether department leaders consistently provide opportunities for knowledge-sharing across various disciplines (*p* = 0.016). Most dentists indicated that department leaders always (45.5%) or often (27.3%) provide opportunities for knowledge-sharing, 18.2% stated seldom and 9.1% never. Responses from pharmacists were more varied, with 56.5% stating seldom, 26.1% often, 8.7% always, and 8.7% never. The majority of AHPs reported seldom (40.0%) or often (36.9%), with fewer indicating always (7.7%) or never (15.3%).

While statistically significant differences were not found across professions for several practices, visible descriptive patterns emerged.

Engaging with the outside community

The results suggest that pharmacists were more likely to report that departments “often” work with the outside community (43.5%), compared with more evenly distributed responses among dentists (27.3%) and AHPs (27.7%;).

Establishing social structures such as communities of practice

Regarding the existence of social structures to bring together health professionals and share ideas across hospital departments, most respondents across all professions selected “seldom” (pharmacists = 47.8%; dentists = 45.5%; AHPs = 44.6%); however, pharmacists also reported slightly higher “never” responses (34.8%).

Fostering a daily knowledge sharing culture

Regarding existence of knowledge-sharing and collaborative culture within their department, dentists reported the highest frequency of “always” (45.5%), whereas pharmacists (52.2% & 26.1%) and AHPs (36.9% & 30.8%) more commonly selected “seldom” or “often”, respectively.

Adopting information systems to support information exchange

Concerning the adoption of information systems for information sharing, the results indicate that dentists reported more frequently selecting “always” (45.5%) compared with pharmacists (8.7%) and AHPs (9.2%).

Encouraging staff returning from training to share new learning

Regarding participation in professional development programs, or the establishment of a long-term departmental knowledge-sharing strategy, there were mixed responses. Most dentists (45.5%) reported “always”, while most pharmacists (56.5%) indicated “seldom”, and AHPs were nearly evenly split between “seldom” (38.5%) and “often” (36.9%).

Establishing long term knowledge sharing

Finally, as shown in Table 2, dentists were more likely to report “always” (45.5%) for the existence of a long-term knowledge-sharing strategy, while pharmacists (47.8%) and AHPs (29.7%) reflected comparatively moderate levels (“seldom”) of agreement.

## 4. Discussion

This study assessed the knowledge-sharing practices among dentists, pharmacists, and AHPs in nine public hospitals in the EC province. The sampled population largely consisted of females, with a significant proportion of participants aged between 21 and 35. AHPs comprised most of the participants, whereas dentists represented the smallest group. This study found statistically significant association between profession and perceptions that the department encourages staff to think from a global perspective. This study also found statistically significant differences observed by profession concerning whether departments encourage staff to seek answers from other teams when solving problems. Regarding the perceptions of whether department leaders consistently provide opportunities for knowledge-sharing across various disciplines, this study found statistically significant differences among professions. Despite observable patterns in the descriptive results, no other statistically significant differences by profession were observed for all other items.

Dentists were the most optimistic about departmental support for global perspectives, inter-team problem-solving, collaboration with external communities, and fostering a knowledge-sharing culture. In contrast, most pharmacists and AHPs felt that global perspectives were seldom encouraged, and they had similar concerns about the opportunities for knowledge-sharing provided by leaders and the presence of long-term strategies. This study also established that social structures supporting knowledge-sharing were generally lacking across all professions, and the adoption of information systems was consistently low, particularly among pharmacists and AHPs. Additionally, dentists’ engagement in development programs was highest, while participation was significantly lower among AHPs and pharmacists.

The demographic profile in this study, characterized by a significant proportion of early-career practitioners and a notable representation of women, suggests a workforce segment poised for sustained professional engagement. This presents opportunities for long-term investment in capacity-building and knowledge-sharing initiatives while highlighting the necessity for gender-responsive policies and leadership pathways to promote equity and enhance healthcare quality [30].

The findings of this study indicate that dentists receive the most encouragement to think globally, while most pharmacists and AHPs do not feel similarly supported. Research on promoting a global perspective in healthcare has revealed varying levels of institutional commitment across different disciplines. Batalden et al. highlight the importance of global thinking in healthcare for improving patient care and enhancing professional adaptability [31]. Similarly, Frenk et al. emphasize that interdisciplinary collaboration is crucial for addressing complex global health challenges [32]. However, this study’s results show that pharmacists and AHPs feel significantly less encouraged to adopt global perspectives than dentists. This finding is consistent with research by Whittington et al. which found that leadership and organizational culture are critical factors in shaping knowledge-sharing behaviours [33]. The observed disparities suggest the presence of systemic barriers within certain professional groups, underscoring the need for targeted interventions to cultivate a more inclusive and globally minded healthcare workforce.

Research consistently highlights the importance of interprofessional collaboration in healthcare, particularly for enhancing patient safety, efficiency, and innovation [34]. However, this study found notable differences in perceptions of teamwork among dentists, pharmacists, and AHPs, indicating that some professional groups may receive more institutional support for collaboration. Studies have demonstrated that healthcare environments that encourage inter-team problem-solving experience have fewer medical errors and higher patient satisfaction [35]. Despite these benefits, siloed structures continue to pose challenges, especially in settings dominated by hierarchical or discipline-specific approaches [36]. To overcome these barriers, it is crucial to implement structured interprofessional education and organizational policies that promote a truly collaborative culture [37].

Research has consistently highlighted the importance of healthcare institutions collaborating with external communities to improve service delivery and public trust. De Weger et al. emphasize that such collaboration ensures healthcare services align with local needs, enhancing patient-centred care [38]. Similarly, Belaid et al. argue that strong community engagement fosters better health outcomes by addressing social determinants of health [39]. However, limited collaboration, as seen in this study, may reflect resource constraints or a lack of structured outreach programs, a challenge also noted by Cometto, Ford, Pfaffman-Zambruni et al. [40], who found that underfunded healthcare systems struggle to sustain meaningful community partnerships. Moreover, Wakerman et al. highlight that weak institutional-community links particularly affect rural and underserved areas, reducing the effectiveness of health interventions [41]. Addressing these gaps requires policy-driven initiatives that prioritise community involvement, adequate funding, and capacity-building efforts to strengthen healthcare-community partnerships.

This study highlights the absence of Communities of Practice, a trend seen in many healthcare settings where these groups are underutilised despite their benefits. Wenger noted that Communities of Practice promote collaboration and learning across different fields, bridging knowledge and practice gaps [42]. Without Communities of Practice, opportunities for improved teamwork and knowledge-sharing are missed. D’Amour et al. found that effective collaboration leads to better care coordination and fewer medical errors [43]. Likewise, West et al. showed that organizations supporting knowledge-sharing through Communities of practice often achieve better patient outcomes and more efficient healthcare [44]. This absence may indicate limited resources or a lack of organizational support for professional development and teamwork.

This study found that a larger proportion of dentists believe their department encourages knowledge-sharing, supporting research that shows teamwork helps professional growth and organisational success. A knowledge-sharing culture is key to innovation and effective healthcare [45]. In contrast, pharmacists and AHPs report less knowledge-sharing, which may hurt their development and morale. Research indicates that a lack of collaboration can lead to lower job satisfaction and poorer patient care [44]. The absence of knowledge-sharing can also hinder the adoption of best practices, impacting care quality [46]. These findings highlight the need for strategies that promote knowledge-sharing across all healthcare fields to improve outcomes for staff and patients.

This study found that most dentists regularly benefit from knowledge-sharing opportunities facilitated by their leaders, supporting existing research highlighting leadership’s significance in fostering continuous learning and professional development. Effective leadership is vital for cultivating environments that promote collaboration and innovation [47]. In contrast, the perception of limited knowledge-sharing among pharmacists and AHPs indicates a leadership gap that can obstruct collaboration and hinder the flow of knowledge within healthcare teams. D’Amour et al. emphasise that leadership is essential for facilitating interprofessional collaboration and nurturing a learning culture [43]. Moreover, research by West et al. demonstrates that clear leadership and active involvement in knowledge-sharing processes directly influence team effectiveness and morale [44]. Insufficient leadership may lead to decreased engagement and missed opportunities for professional growth [44]. Ultimately, these findings underscore the critical role of leadership in enhancing collaboration and ensuring that all staff members benefit from knowledge-sharing practices.

This study reveals that dentists are more likely to use information systems for knowledge-sharing between departments than pharmacists and AHPs, highlighting the increasing significance of technology in healthcare. Sittig and Singh emphasize that effective information systems are vital for quick data sharing and improved decision-making [48]. Conversely, pharmacists and AHPs have lower adoption rates of these systems, which can disrupt the flow of critical information. Health information technology enhances patient outcomes and operational efficiency [49]. Improper use of these systems can result in communication problems and errors that negatively impact patient care [50]. These findings suggest that healthcare organizations should invest in information systems and provide training for staff to maximize patient care and improve overall operations.

Access to continuing professional development (CPD) programmes varies considerably across health disciplines in South Africa. While CPD is mandatory for many categories of registered health practitioners under the Health Professions Council of South Africa (HPCSA), including doctors, dentists and allied professions [51], empirical evidence reveals significant barriers to participation for some professions. A survey of radiographers in KwaZulu-Natal, for instance, found low CPD uptake due to funding shortages, time constraints, limited employer support, and scarce accredited training opportunities [52]. Similarly, compliance with CPD requirements among dental technicians remains low, reflecting structural and resource limitations [53]. These disparities likely explain the uneven distribution of clinical-governance knowledge observed in our study. Therefore, it is critical to consider differential access to CPD when our findings are interpreted and when designing capacity-building interventions. This study found that dentists to be more likely to participate in professional development programs than pharmacists and AHPs. Dentists may be more motivated to engage in continuing CPD due to their greater clinical autonomy, which allows them to maintain competence, protect their independent practice, and meet regulatory standards [54]. In contrast, pharmacists’ clinical decision-making autonomy is restricted by organizational structures, regulatory frameworks, and the necessity to align with prescribers’ decisions [55,56]. As a result, pharmacists have less control over patient management, which may diminish their urgency to pursue CPD for independent decision-making. Similarly, AHPs often work in multidisciplinary teams and operate within referral systems and shared decision-making frameworks, which can further reduce their individual clinical accountability compared to dentists [55]. This aligns with existing literature emphasizing the value of CPD in healthcare. Kolb notes that training enhances learning and skill acquisition, which are vital for improving healthcare outcomes [57]. In contrast, lower participation rates among pharmacists and AHPs suggest barriers to accessing these opportunities. A study by Shiri et al. indicates that unequal access to training can hinder professional growth and workplace satisfaction [58]. Limited career development options are said to be associated with lower job satisfaction and retention in some healthcare professions [59]. Our study findings highlight the need for equitable access to development programs, enabling all healthcare professionals to enhance their skills and improve overall service delivery.

This study also found that the belief in a long-term knowledge-sharing strategy among dentists differs from the views of pharmacists and AHPs. This contrast reflects broader trends in healthcare organizations. The absence of a clear strategic focus and knowledge-sharing can hinder innovation and adaptability to changing healthcare demands [45]. Bontis et al. found that strong knowledge-sharing strategies promote innovation and responsiveness, while their absence can lead to fragmentation and inefficiencies, especially for pharmacists and AHPs trying to collaborate [60]. Organizations lacking such strategies may struggle to retain expertise and maintain a competitive edge, ultimately affecting the quality of care [61].

These findings provide important insights about how healthcare professionals share knowledge, work together, and lead in hospitals. Thus, these insights can help improve healthcare systems, workplace culture, and public health outcomes. Furthermore, when healthcare professionals collaborate and share information effectively, they can lower medical errors, improve care coordination, and achieve better treatment results. Fixing communication problems between departments can help reduce service delays and better use of resources [43]. These findings also encourage hospital management to adopt strategies that foster teamwork, such as creating Communities of practice or cross-functional teams [42]. Also, this study highlights the need for leadership in knowledge-sharing, suggesting that hospitals should create training programs for healthcare managers and department heads.

This study addresses a gap in research by focusing on knowledge-sharing practices among AHPs, dentists, and pharmacists, who are often underrepresented in health systems research. By examining the knowledge-sharing practices, the study aims to identify ways to enhance collaboration and improve service delivery, which is essential for optimizing patient care, particularly in resource-constrained settings. However, the researchers acknowledge the fact that the results from the dentists are largely different from pharmacists and AHP might be attributed to smaller representation of dentists in the population. The cross-sectional design used in this study does not capture knowledge-sharing practices over an extended period. Additionally, the findings may not be generalizable to other provinces, private settings, or larger national contexts. Despite these limitations, the findings of this study highlight the need to strengthen leadership-driven opportunities for interdisciplinary knowledge-sharing and to develop targeted interventions to address specific gaps between professions. Additionally, the authors recommend further research that tracks how knowledge-sharing practices evolve over time and in response to interventions or policy changes.

## 5. Conclusions

This study provides important insights into knowledge-sharing practices among dentists, pharmacists, and allied health professionals working in public hospitals in the Eastern Cape. While many aspects of knowledge sharing were perceived similarly across professions, statistically significant differences emerged in staff experiences related to global thinking, cross-team problem-solving, and leadership support for interdisciplinary exchange. These findings indicate that professional identity and role expectations significantly influence individual engagement with organizational learning processes, even within a shared institutional context. Conversely, the lack of differences in most areas suggests that overarching organizational structures and cultures exert a unifying influence on knowledge-sharing behaviours. Enhancing these systems while addressing profession-specific gaps presents a promising avenue for improving collaborative learning and service delivery across disciplines.

## Figures and Tables

**Table 1 ijerph-23-00066-t001:** Demographic characteristics of dentists and pharmacists [29].

Variable	Dentist (n = 11)	Pharmacist (n = 23)	AHPs (n = 65)	Total (N = 99)	
Age, years; Median (IQR)				32 (13)	
				Min	Max
				21	62
Age Groups; n (%)					
21–35 years	8 (12.7)	11 (17.5)	44 (69.8)	63 (100)	
36–45 years	2 (9.5)	9 (42.9)	10 (47.6)	21 (100)	
46–55 years	0 (0.0)	2 (20.0)	8 (80.0)	10 (100)	
>55 years	1 (20.0)	1 (20.0)	3 (60.0)	5 (100)	
Sex ^#^; n (%)					
Female	7 (9.2)	15 (19.7)	54 (71.1)	76 (100)	
Male	4 (18.2)	7 (31.8)	11 (50.0)	22 (100)	
Hospital; n (%)					
Frere hospital	3 (13.0)	6 (26.1)	14 (60.9)	23 (100)	
Cecilia Makiwane hospital	3 (13.6)	5 (22.7)	14 (63.6)	22 (100)	
Mthatha Regional hospital	1 (5.6)	3 (16.7)	14 (77.8)	18 (100)	
Frontier hospital	1 (8.3)	3 (25.0)	8 (66.7)	12 (100)	
Madzikane KaZulu hospital	1 (14.3)	2 (28.6)	4 (57.1)	7 (100)	
Dr. Malizo Mpehle hospital	1 (14.3)	2 (28.6)	4 (57.1)	7 (100)	
All Saints hospital	1 (25.0)	1 (25.0)	2 (50.0)	4 (100)	
Butterworth hospital	0 (0.0)	1 (33.3)	2 (66.7)	3 (100)	
Bisho hospital	0 (0.0)	0 (0.0)	3 (100.0)	3 (100)	

Notes: AHPs = Allied Health Professionals; ^#^ = n = 98.

**Table 2 ijerph-23-00066-t002:** Systems encourage knowledge-sharing and theories espoused by department members.

	Never	Seldom	Often	Always	Total	χ^2^ *p*-Value
The department encourages people to think from a global perspective; n (%)						
Dentist	1 (9.1)	3 (27.3)	3 (27.3)	4 (36.4)	11	0.017
Pharmacist	6 (26.1)	13 (56.5)	3 (13.0)	1 (4.3)	23	
AHPs	6 (9.2)	35 (53.9)	19 (29.2)	5 (7.7)	65	
The department encourages people to get answers from across teams when solving problems; n (%)						**χ^2^ *p*** **-value**
Dentist	0 (0.0)	2 (18.2)	4 (36.4)	5 (45.5)	11	0.020
Pharmacist	3 (13.0)	13 (56.5)	5 (21.7)	2 (8.7)	23	
AHPs	4(6.2)	29 (44.6)	25 (38.5)	7 (10.8)	65	
The department works together with the outside community to meet mutual needs; n (%)						**χ^2^ *p*** **-value**
Dentist	1 (9.1)	3 (27.3)	3 (27.3)	4 (36.4)	11	0.089
Pharmacist	4 (17.4)	9 (39.1)	10 (43.5)	0 (0.0)	23	
AHPs	6 (9.2)	32 (48.4)	18 (27.7)	9 (13.9)	65	
Social structures such as communities of practice are in place to bring together health professionals and share ideas across hospital departments; n (%)						**χ^2^ *p*** **-value**
Dentist	2 (18.2)	5 (45.5)	2 (18.2)	2 (18.2)	11	0.488
Pharmacist	8 (34.8)	11 (47.8)	3 (13.0)	1 (4.3)	23	
AHPs	12 (18.5)	29 (44.6)	18 (27.7)	6 (9.2)	65	
The department has built a knowledge-sharing and collaborative culture that health professionals practice daily; n (%)						**χ^2^ *p*** **-value**
Dentist	1 (9.1)	2 (18.2)	3 (27.3)	5 (45.5)	11	0.194
Pharmacist	3 (13.0)	12 (52.2)	6 (26.1)	2 (8.7)	23	
AHPs	14 (21.5)	24 (36.9)	20 (30.8)	7 (10.8)	65	
Department leaders provide constant opportunities for knowledge-sharing; n (%)						**χ^2^ *p*** **-value**
Dentist	1 (9.1)	2 (18.2)	3 (27.3)	5 (45.5)	11	0.016
Pharmacist	2 (8.7)	13 (56.5)	6 (26.1)	2 (8.7)	23	
AHPs	10 (15.3)	26 (40.0)	24 (36.9)	5 (7.7)	65	
The department has adopted information systems to make it easy to share information; n (%)						**χ^2^ *p*** **-value**
Dentist	1 (9.1)	2 (18.2)	3 (27.3)	5 (45.5)	11	0.099
Pharmacist	2 (8.7)	13 (56.5)	6 (26.1)	2 (8.7)	23	
AHPs	10 (15.3)	25 (38.5)	24 (36.9)	6 (9.2)	65	
The department encourages health professionals participating in development/training programs to engage with others to share what they have learned; n (%)						**χ^2^ *p*** **-value**
Dentist	1 (9.1)	2 (18.2)	3 (27.3)	5 (45.5)	11	0.147
Pharmacist	2 (8.7)	13 (56.5)	6 (26.1)	2 (8.7)	23	
AHPs	10 (15.3)	26 (40.0)	24 (36.9)	5 (7.8)	65	
There is a long-term knowledge sharing-strategy in my department; n (%)						**χ^2^ *p*** **-value**
Dentist	2 (18.2)	2 (18.2)	2 (18.2)	5 (45.5)	11	0.069
Pharmacist	6 (26.1)	11 (47.8)	5 (21.7)	1 (4.4)	23	
AHPs	14 (21.5)	19 (29.7)	22 (33.9)	10 (15.3)	65	

Notes: AHPs = Allied health professional

## Data Availability

The raw data supporting the conclusions of this article will be made available by the authors on request.

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
