# Peer review of "Knowledge-Sharing Practices Among Dentists, Pharmacists, and Allied Health Professionals: A Cross-Sectional Study in Eastern Cape Public Hospitals, South Africa"

_ijerph, 2025, doi:10.3390/ijerph23010066_

Round 1
Reviewer 1 Report
Comments and Suggestions for Authors
This manuscript presents a cross-sectional survey assessing knowledge-sharing practices among dentists, pharmacists, and allied health professionals (AHPs) in nine public hospitals in Eastern Cape, South Africa. The study explores disparities in collaborative culture, systems, and leadership engagement among different health professional groups.
1. Findings of your study may not be generalizable to other provinces, private settings, or larger national contexts. please edit it.
2.Please provide a more detailed account of the questionnaire development and validation processes to enhance the methodological rigor of the study.
3.How you choose the hospitals?
4. The section of results is simple and shows solely descriptive statistics. More advanced analyses (e.g., inferential statistics, multivariable approaches) could help clarify associations between demographics and knowledge-sharing practices
Comments on the Quality of English LanguageOverall, the use of the English language is adequate for publication, with only minor edits recommended to enhance clarity and flow.
Author Response
General comment: This manuscript presents a cross-sectional survey assessing knowledge-sharing practices among dentists, pharmacists, and allied health professionals in nine public hospitals in Eastern Cape, South Africa. The study explores disparities in collaborative culture, systems, and leadership engagement among different health professional groups.
Reviewer comment: Findings of your study may not be generalizable to other provinces, private settings, or larger national contexts. please edit it.
Author response: Thank you for this comment. The statement has been edited as per the reviewer advice. Lines: 452-453.
Reviewer comment: Please provide a more detailed account of the questionnaire development and validation processes to enhance the methodological rigor of the study.
Author response: Thank you for this comment. The details about the development and validation of the questionnaire have been added in Lines: 174-182. The questionnaire was developed through literature review and established theoretical frameworks to ensure conceptual coverage. Content validity was assessed by two senior academic researchers, an organizational behaviour specialist and a public health medicine specialist, who evaluated each item for relevance, clarity, and comprehensiveness. Following content validation, the questionnaire was piloted with 10 individuals from the target population. Internal consistency of the nine-item knowledge-sharing scale was assessed using Cronbach's alpha coefficient, with values ≥0.70 considered acceptable for reliability (30,31). The scale demonstrated excellent reliability (α = 0.88), exceeding the recommended threshold of 0.70 for research instruments.
Reviewer comment: How you choose the hospitals?
Response: Thank you for this comment. The details about the selection of the hospitals are specified in Lines: 156-157. “Nine hospitals were purposively sampled based on their involvement in training Walter Sisulu University’s students.”
Reviewer comment: The section of results is simple and shows solely descriptive statistics. More advanced analyses (e.g., inferential statistics, multivariable approaches) could help clarify associations between demographics and knowledge-sharing practices.
Response: Thank you for this comment. The results section has been revised see Tables 1 & 2. Specifically, in Table 2 inferential statistics (Chi-square test) have been used in Lines:
Comments on the Quality of English Language: Overall, the use of the English language is adequate for publication, with only minor edits recommended to enhance clarity and flow.
Response: Thank you for this comment. Some aspects of the manuscripts have been revised to enhance clarity and flow.
Reviewer 2 Report
Comments and Suggestions for Authors
annexed

can be improved for clearer presentation
Author Response
General reviewer comment: This is an important topic to discuss as knowledge sharing to the optimum levels will improve the treatment outcomes in any health concern.
Introduction: is comprehensive with identification of benefits, limitations and barriers with sufficient explanation. However, it needs to be re-organized for better presentation.
Reviewer comment: E.g. 1
Despite the acknowledgement of knowledge-sharing among different health systems globally, it remains poorly practised (5). In paragraph 3
Even though the importance of knowledge and experience-sharing practices are mentioned by various studies, they are poorly practised in hospitals of resource-scared countries (3). paragraph 6 both same facts. But written haphazard
Author response: Thank you for this comment. Paragraph 3 has been revised to incorporate the statement in paragraph 6: Lines 57-59. The revised statement “Despite the acknowledgement of knowledge-sharing among different health systems globally, it remains poorly practised (5), especially in hospitals of resource-scared countries (3).”
- Reviewer comment: g. 2
Knowledge-sharing offers inherent benefits and can occur at individual and organisational levels (3). Paragraph 2
Existing research demonstrates that hospitals with strong learning cultures and effective knowledge-sharing systems can deliver high-quality care and innovate in response to emerging health challenges (18). Paragraph 5: Benefits described but need to remove repetitions and re organization
Author response: Thank you for this comment. The statement in Paragraph 2 has been moved and incorporated into paragraph 5: Lines 87-90. The revised paragraph reads as follows: “Knowledge-sharing at individual and organizational levels yields inherent benefits (3). Research demonstrates that hospitals with strong learning cultures and effective knowledge-sharing systems can deliver high-quality care and respond innovatively to emerging health challenges (18)…”
- Reviewer comment: g. 3
A study conducted in Taiwan revealed that support from top management, organizational rewards, and technological resources are linked to knowledge sharing practices (5). Paragraph 5
These knowledge-sharing practices can be attributed to education level, experience, awareness, willingness, teamwork, availability of health information resources, communication mechanisms, internet access, and computer literacy (22). Paragraph 6
Author response: Thank you for this comment. The statement in Paragraph 5 has been revised to incorporate the statement from paragraph 6: Lines 94-101. The revised paragraph reads as follows: “A study conducted in Taiwan revealed that support from top management, organizational rewards, and technological resources are linked to knowledge-sharing practices (5). Other factors that can influence knowledge-sharing practices include education level, experience, awareness, willingness, teamwork, availability of health information resources, communication mechanisms, internet access, and computer literacy (22). Another study in Ethiopia suggests that factors that negatively affect knowledge-sharing practices include individual factors and organisational and resource-related issues (3).”
Reviewer comment: Justification for the study such as Healthcare delivery is becoming complex, requiring professionals from diverse disciplines to collaborate and make timely clinical decisions. When knowledge is not shared adequately, the effects such as it can lead to fragmented care, duplication of work, preventable errors, and suboptimal patient outcomes. to be included into justification to highlight the importance of your study with regard to the region of South Africa especially, the fact that is described as: Healthcare services are not easily accessible in the EC province. The statement: Healthcare services are not easily accessible in the EC province (19). The healthcare system in the EC province is characterised by inequity of access between rural and urban populations. Tertiary or quaternary health facilities are located in urban areas. Thus, patients from rural areas who cannot afford private care are forced to travel long distances to seek healthcare in urban-based tertiary or quaternary health facilities (29) in study setting could be used for justification.
Author response: Thank you for the comment. We have considered your feedback/input and the justification for this study has been improved in Liens: 116-127.
Reviewer comment: Methods section is well described. Ethical principles are well adopted.
Author response: Thank you for this comment.
Reviewer comment: Results. There are interesting and significant findings. Well presented in figures and tables.
Author response: Thank you for this comment.
Reviewer comment: Discussion: The results from the dentists are largely different from pharmacists and AHP. This could be attributed to smaller representation of dentists in the population. The facts should be discussed as a limitation in the study. Further, dentists working with own decisions than AHP and pharmacists may be another factor for the results which need to be discussed too.
Author response: Thank you for this comment. This comment has been addressed in Lines: 404-414.
Reviewer comment: Have you considered proportion of CPD programs available for different disciplines in South Africa. That should be discussed.
Author response: Thank you for this comment. This comment has been addressed in Lines: 419-430. “Access to continuing professional development (CPD) programmes and therefore opportunities to learn about clinical-governance protocols and tools varies considerably across health disciplines in South Africa. While CPD is mandatory for many categories of registered health practitioners under the Health Professions Council of South Africa (HPCSA), including doctors, dentists and allied professions (7), empirical evidence reveals significant barriers to participation for some professions. A survey of radiographers in KwaZulu-Natal, for instance, found low CPD uptake due to funding shortages, time constraints, limited employer support, and scarce accredited training opportunities (8). Similarly, compliance with CPD requirements among dental technicians remains low, reflecting structural and resource limitations (9). These disparities likely explain the uneven distribution of clinical-governance knowledge observed in our study. Differential access to CPD should therefore be considered when interpreting our findings and when designing capacity-building interventions.”
Reviewer comment: “The research also shows that hospitals need to work more closely with outside communities”. Describe how this statement relates to the study objectives.
Author response: Thank you for this comment. This statement has been removed since it does not take away any meaning weight from the paragraph.
Reviewer comment: Addition of the recommendations drawn out of the findings would add more value to the study. Such as a recommendation can be more emphasis on combined programs to update on sharing knowledge for all professionals to improve collaborations
Author response: Thank you for this comment. Recommendations have been added in line with the findings. Lines: 453-456 as follows: Despite these limitations, the findings of this study highlight the need to strengthen leadership-driven opportunities for interdisciplinary knowledge-sharing and to develop targeted interventions to address specific gaps between professions.
Reviewer 3 Report
Comments and Suggestions for Authors
The manuscript, “Knowledge-sharing practices among dentists, pharmacists, and allied health professionals: a cross-sectional study in Eastern Cape public hospitals, South Africa,” addresses an important and often underexplored aspect of interprofessional collaboration in resource-constrained health systems. The topic is relevant and potentially impactful for improving healthcare delivery and professional development.
However, several major revisions are needed to strengthen clarity, context, and methodological rigor.
- Introduction
The Introduction lacks sufficient contextual information about the organization of the South African health system and how different professional groups function within it. Readers unfamiliar with this setting may struggle to understand the structural factors influencing knowledge-sharing. Some of this already appears in the Study Setting section and could be moved or expanded in the Introduction.
- Statistical Analysis
The statistical analysis is currently too limited. Descriptive statistics alone cannot support claims about differences or relationships between professional groups. I recommend including more statisthycal methods for improving the clarity of the manuscript.
- Discussion
Line 324: The reference "(41)" is incomplete; the authors' name should be included
Line 341: The phrase “significant proportion” needs clarification. Does this refer to statistical significance or simply a larger percentage numerically? The wording should be adjusted accordingly to avoid misinterpretation.
The manuscript has potential, but important improvements are needed as mentioned above. Addressing these points would substantially strengthen the study and its contribution to the field.
Author Response
General reviewer comment: The manuscript, “Knowledge-sharing practices among dentists, pharmacists, and allied health professionals: a cross-sectional study in Eastern Cape public hospitals, South Africa,” addresses an important and often underexplored aspect of interprofessional collaboration in resource-constrained health systems. The topic is relevant and potentially impactful for improving healthcare delivery and professional development.
However, several major revisions are needed to strengthen clarity, context, and methodological rigor.
- Introduction
Reviewer comment: The Introduction lacks sufficient contextual information about the organization of the South African health system and how different professional groups function within it. Readers unfamiliar with this setting may struggle to understand the structural factors influencing knowledge-sharing. Some of this already appears in the Study Setting section and could be moved or expanded in the Introduction.
Author response: Thank you for the comment. the Introduction has been improved to incorporate additional contextual information.
- Statistical Analysis
The statistical analysis is currently too limited. Descriptive statistics alone cannot support claims about differences or relationships between professional groups. I recommend including more statistical methods for improving the clarity of the manuscript.
Response: Thank you for this comment. The results section has been revised see Tables 1 & 2. Specifically, in Table 2 inferential statistics (Chi-square test) have been used.
- Discussion:
Reviewer comment: Line 324: The reference "(41)" is incomplete; the authors' name should be included
Author response: Thank you for the comment. The citation has been updated in Line 354-355 as follows: “Cometto, Ford, Pfaffman-Zambruni et al.,”.
Reviewer comment: Line 341: The phrase “significant proportion” needs clarification. Does this refer to statistical significance or simply a larger percentage numerically? The wording should be adjusted accordingly to avoid misinterpretation.
Author response: Thank you for the comment. The phrase “…significant proportion…” has been replaced with “…larger proportion…”in Line 371.
General reviewer comment: The manuscript has potential, but important improvements are needed as mentioned above. Addressing these points would substantially strengthen the study and its contribution to the field.
Reviewer 4 Report
Comments and Suggestions for Authors
The authors raised the important topic of cooperation and knowledge sharing among specialists in various fields of medicine. They should add some information though.
Introduction:
1.What methods of knowledge sharing did they mention? Were they practiced among the respondents?
- line 88 - What individual, organizational and technological factors might hinder or facilitate effective knowledge sharing and a learning cultur?
Materials and Methods
What were the excluding and including criteria to participate in the study?
Discussion
line 288- Why did the demographic composition of respondents in this study present opportunities for long-term professional development and knowledge-sharing initiatives to enhance healthcare quality?
Conclusions
What was the aim of the study? The conclusions do not correspond to the aim.
Author Response
The authors raised the important topic of cooperation and knowledge sharing among specialists in various fields of medicine. They should add some information though.
Introduction:
Comment: What methods of knowledge sharing did they mention? Were they practiced among the respondents?
Response: This comment is not clear.
Reviewer comment: line 88 - What individual, organizational and technological factors might hinder or facilitate effective knowledge sharing and a learning culture?
Author response: Thank you for this comment. We have clarified the rationale in the revised manuscript: Lines: 91-94. “This comment has been addressed as follows: Various individual (e.g., self-efficacy, motivation, trust), organisational (e.g., leadership support, collaborative culture, workload) and technological factors (e.g., usability of ICT systems, access to digital tools) can hinder or facilitate effective knowledge-sharing and learning culture (1,20).”
Materials and Methods
Reviewer comment: What were the excluding and including criteria to participate in the study?
Response: Thank you for this comment. We have clarified excluding and including criteria to participate in the study: Lines: 163-172. Several considerations were made before participants were included in this study. Participants were eligible if they were registered dentists, pharmacists, or allied health professionals currently employed in the study hospitals and providing clinical or direct patient-related services; were aged 18 years or older; have been employed at the study hospital for at least six months to ensure familiarity with local practices and were willing to provide written informed consent. While administrative staff with no clinical duties, undergraduate students, staff on extended leave (more than 3 months) during data collection, and those with survey responses with excessive missing data (> 20% on key items) were not considered for inclusion in this study.
Discussion
Reviewer comment: Line 288- Why did the demographic composition of respondents in this study present opportunities for long-term professional development and knowledge-sharing initiatives to enhance healthcare quality?
Response: Thank you for this comment. We have clarified the rationale in the revised manuscript: Lines: 317-322. “The demographic profile, characterised by a significant proportion of early-career practitioners and a notable representation of women, suggests a workforce segment poised for sustained professional engagement. This presents opportunities for long-term investment in capacity-building and knowledge-sharing initiatives while highlighting the necessity for gender-responsive policies and leadership pathways to promote equity and enhance healthcare quality.”
Conclusions
Reviewer comment: What was the aim of the study? The conclusions do not correspond to the aim.
Author response: Thank you for this comment. The conclusion has been revised in Lines: 460-471. “This study provides important insights into knowledge‑sharing practices among dentists, pharmacists, and allied health professionals working in public hospitals in the Eastern Cape province. While many aspects of knowledge sharing were perceived similarly across professions, statistically significant differences emerged in staff experiences related to global thinking, cross-team problem-solving, and leadership support for interdisciplinary exchange. These findings indicate that professional identity and role expectations significantly influence individual engagement with organizational learning processes, even within a shared institutional context. Conversely, the lack of differences in most areas suggests that overarching organizational structures and cultures exert a unifying influence on knowledge-sharing behaviours. Enhancing these systems while addressing profession-specific gaps presents a promising avenue for improving collaborative learning and service delivery across disciplines.
Round 2
Reviewer 3 Report
Comments and Suggestions for Authors
All of the suggestions were implemented into the text.
Author Response
I am not sure what I was meant to address here. However, I saw a comment suggesting that "Tables can be improved". We have improved the formatting of the tables.